# Loss of Peter Pan (PPAN) Affects Mitochondrial Homeostasis and Autophagic Flux

**DOI:** 10.3390/cells8080894

**Published:** 2019-08-14

**Authors:** David P. Dannheisig, Eileen Beck, Enrico Calzia, Paul Walther, Christian Behrends, Astrid S. Pfister

**Affiliations:** 1Institute of Biochemistry and Molecular Biology, Ulm University, D-89081 Ulm, Germany; 2Institute of Anesthesiological Pathophysiology and Process Development, Ulm University, D-89081 Ulm, Germany; 3Central Facility for Electron Microscopy, Ulm University, D-89081 Ulm, Germany; 4Munich Cluster for Systems Neurology, Medical Faculty, Ludwig-Maximilians-University München, D-81377 Munich, Germany

**Keywords:** apoptosis, autophagy, mitochondria, mitophagy, nucleolus, nucleolar stress, Parkin, PPAN, ribosome, Wnt target

## Abstract

Nucleolar stress is a cellular response to inhibition of ribosome biogenesis or nucleolar disruption leading to cell cycle arrest and/or apoptosis. Emerging evidence points to a tight connection between nucleolar stress and autophagy as a mechanism underlying various diseases such as neurodegeneration and treatment of cancer. Peter Pan (PPAN) functions as a key regulator of ribosome biogenesis. We previously showed that human PPAN localizes to nucleoli and mitochondria and that PPAN knockdown triggers a p53-independent nucleolar stress response culminating in mitochondrial apoptosis. Here, we demonstrate a novel role of PPAN in the regulation of mitochondrial homeostasis and autophagy. Our present study characterizes PPAN as a factor required for maintaining mitochondrial integrity and respiration-coupled ATP production. PPAN interacts with cardiolipin, a lipid of the inner mitochondrial membrane. Down-regulation of PPAN enhances autophagic flux in cancer cells. PPAN knockdown promotes recruitment of the E3-ubiquitin ligase Parkin to damaged mitochondria. Moreover, we provide evidence that PPAN knockdown decreases mitochondrial mass in Parkin-expressing cells. In summary, our study uncovers that PPAN knockdown is linked to mitochondrial damage and stimulates autophagy.

## 1. Introduction

Nucleoli represent specific sub-nuclear compartments, which are essential for ribosome biogenesis. Moreover, they function as central hub in the cellular stress response by inducing so-called nucleolar stress, an impairment of nucleolar structure and/or function [1]. As a consequence, the stressed cells respond by stabilization of the tumor suppressor p53 and undergo cell cycle arrest and/or apoptosis [2]. Over the past few years, various p53-dependent and -independent mechanisms have been added to the increasing list of nucleolar stress response pathways [3]. It has become apparent that a tight regulation of the nucleolar stress response is mandatory for a healthy cell.

Induction of nucleolar stress is applied as novel and promising anti-cancer therapy to trigger apoptosis of the cancerous cells. At the same time, nucleolar stress is tightly linked to various aging-related diseases, such as neurodegeneration, which is characterized by cell death of neuronal cells and accumulation of defective mitochondria [4,5,6,7]. For instance, nucleolar stress has been shown to precede neurodegeneration in a mouse model for Huntington’s disease with activated autophagy, and is also observed in patients with so-called ribosomopathies [5,8]. In line with this, nucleolar factors are increasingly uncovered to participate in the regulation of autophagic pathways [9,10,11]. Autophagy research has gained tremendous attention over the last few years, as this pathway is involved in aging and neurodegeneration as well as cancer [12,13,14]. Autophagy is an evolutionary conserved process responsible for degradation and recycling of damaged proteins or organelles [15]. Autophagy plays a protective and energy-preserving role for the cell. However, a proper balance of autophagy activation and inhibition has to be ensured as a prerequisite for normal cellular function. Central to autophagy is the formation of double-membranous autophagosomes, which later fuse with lysosomes to allow degradation of the engulfed cargo by acidic hydrolases [16]. Under basal growth conditions, microtubule associated protein 1A/1B light chain 3 (MAP1LC31A/B, LC3) exists in a soluble form (LC3-I). Following autophagy induction, for instance by nutrient deprivation or other types of stress, LC3 is conjugated to the lipid phosphatidylethanolamine by several autophagy-related (ATG) proteins and is subsequently integrated into autophagosomal membranes as lipidated LC3-II. However, accumulation of LC3-II can result from either impaired turnover or activation of autophagosomal degradation. To discriminate between both scenarios, it is indispensable to perform so-called autophagic flux experiments, in which lysosomal degradation is blocked [17,18]. 

Specific cargo can be cleared by selective autophagic pathways mostly involving ubiquitination of the particular cargo [19]. The pathways were named according to their type of cargo, e.g., mitophagy for mitochondria [20,21,22]. Mitochondria represent the powerhouse of the cell, they sense threshold of cellular stress and regulate apoptosis. Mitophagy therefore guarantees mitochondrial homeostasis. Parkin and PINK1 (tumor suppressor phosphatase and tensin homologue (PTEN)-induced putative kinase 1) are essential regulators of mitophagy and both have been found to be mutated in patients with Parkinson’s disease [23]. Parkin functions as an E3-ubiquitin ligase, which is recruited to damaged mitochondria [24]. Parkin is essential for ligation of ubiquitin marks to defective mitochondria [13,25] and depends on the proper function of PINK1 [26], an ubiquitin kinase located at the mitochondrial outer membrane (MOM) [27]. Ubiquitinated mitochondria are then recognized by mitophagy receptors and are in turn engulfed by autophagosomes.

Mitophagy and apoptosis engage similar upstream mechanisms [28,29]. Induction of mitochondrial apoptosis, for instance, is initiated by activation and translocation of pro-apoptotic Bcl-2 family members such as BAX/BAK from the cytosol to the MOM [30]. This induces permeabilization of the MOM (MOMP), depolarization and release of cytochrome *c* from the inter-membrane space (IMS) into the cytosol [31]. As a consequence, caspases are activated and transduce the death signal [32]. Note that cardiolipin—a phospholipid of mitochondrial membranes—serves as a platform for pro-apoptotic processes such as BAX-dependent permeabilization of the mitochondrial outer membrane [33,34].

Peter Pan (PPAN) was initially identified in *Drosophila melanogaster* and shown to be highly conserved and essential for maintaining growth and survival [35]. Expression of PPAN is induced in mouse and *Xenopus laevis* by the Wnt signaling pathway [36,37]. The yeast counterpart of PPAN, termed Ssf1, was shown to be a nucleolar ribosome biogenesis factor required for maturation of the large ribosomal subunit [38,39]. 

We recently uncovered that PPAN localizes not only to nucleoli, but also to mitochondria and that PPAN can shuttle between the nucleus and the cytoplasm in response to nucleolar stress and apoptosis induction [40]. We showed that knockdown of human PPAN induces nucleolar stress and triggers mitochondrial apoptosis as observed by induction of mitochondrial depolarization, stabilization of the pro-apoptotic factor BAX (Bcl-2-associated X protein) and release of cytochrome *c* into the cytosol [40]. Strikingly, these effects were independent of stabilization of the tumor suppressor p53, demonstrating that PPAN orchestrates a novel p53-independent nucleolar stress response [40]. We also found that PPAN knockdown is linked to cell cycle defects and that PPAN depletion induces p53/p21-independent, but caspase-dependent H2A.X phosphorylation [41]. Interestingly, apoptosis induction was prominent in cancer cells, but not detectable in human fibroblasts [41]. 

To better understand the role of PPAN in mitochondria, we started to characterize the domains that are necessary and sufficient to target PPAN to mitochondria. We found that the C-terminal half of PPAN (comprising amino acids 287–473) accumulates specifically in mitochondria and revealed the presence of a nuclear export signal (NES), which was essential for mitochondrial targeting [40]. In contrast, the N-terminal part (2–286) of PPAN localized to the nucleoli and nucleus, as it contains the rRNA interacting domains (Brix and σ70). Moreover, mitochondrial prediction algorithms suggested the presence of a N-terminal, cleavable pre-sequence, which might mediate translocation of PPAN into mitochondria [40]. 

So far, the function of mitochondrial PPAN as well as a contribution in autophagy remained elusive. In this study, we uncover that PPAN depletion is associated with mitochondrial damage and induction of autophagy. We propose that the pro-autophagic program serves as initial surveillance mechanism that precedes apoptosis in PPAN knockdown cells.

## 2. Materials and Methods

### 2.1. Plasmids and siRNAs

The PPAN and control siRNAs were characterized previously [40]. Sequences are: si control: GCUACCUGUUCCAUGGCCA si PPAN-B: GGACGAUGAUGAACAGGAA Custom siRNAs were obtained from Thermo Scientific and si PPAN-A was from Qiagen (FlexiTube siRNA #SI00125545). 

pEGFP-Parkin (Addgene plasmid #45875) was a kind gift from Edward Fon and obtained via Addgene. GST-PPAN and GST-PPAN constructs encoding amino acids (2–286) and (287–473) were cloned by PCR from FLAG-PPAN [40] and were inserted into a modified pGEX-4T3 (GE Healthcare). 

### 2.2. Antibodies, Dyes and Drugs

Commercial antibodies were purchased from following companies. Cell Signaling: COX IV (3E11), HSP60 (D307), GAPDH (1410C), LC3A/B (#4108), PARP (#9542), Ubiquitin (P4D1); nanoTools: LC3 (#0321-100); Proteintech: PPAN (#11006-1-AP); BD: TIM23 (#611222); Santa Cruz: LAMP2 (H4B3), TOM20 (FL-145); Sigma: GST (G1160, clone2); Abcam: p62 (EPR4844). Secondary antibodies were IRDye conjugates 800CW and 680CW (Li-COR) for Western blotting or rabbit Alexa 488 (Life Technologies), Alexa 488, Alexa 594, Cy2 and Cy3 conjugates (Dianova) for immunofluorescence staining. DAPI mounting medium was purchased from Dianova. Proteinase K (PK) was from NEB, BafA1 (cat. no. 54645) from Cell Signaling, Oligomycin (cat. no. 1404-19-9) was from Calbiochem, Antimycin A (cat. no. ALX-380-075-M010) and staurosporine (cat. no. ALX-380-014-L250) were from Enzo Life Sciences, CQ (cat. no. C66289) and CCCP (cat. no. C2759) were from Sigma. 

### 2.3. Cell Culture, Transfections and Drug Treatments

Cells were grown in DMEM (high glucose) supplemented with penicillin and streptomycin, 10% FCS and cultured at 37 °C with 5% CO_2_. Cells were used in low passage numbers and were routinely tested for mycoplasma absence (GATC). HEK293A GFP-LC3 cells (ECACC 14050801) from Sigma [42] were cultured in medium supplemented with 0.4 mg/mL G418 (Sigma). HCT116 and U2OS cells were purchased from ATCC. 

Cells seeded in 6-wells were transfected with 45 nM siRNAs for 48 h using Oligofectamine or were co-transfected with 40 nM siRNAs along with 1 µg plasmid using Lipofectamine 2000 (Life Technologies). HEK293A cells were transfected with Oligofectamine or Lipofectamine 2000 with 45 nM siRNAs according to the manufacturer’s instructions.

24 h drug treatments were started 24 h post transfection and short-term treatments 48 h post-transfection. The concentrations used were: CQ (50 µM), BafA1 (100 nM), CCCP (1, 10 or 20 µM, as indicated), O/A = 10 µM Oligomycin plus 5 µM Antimycin A, STS (330 nM). EBSS (Earle’s balanced salt solution) medium was purchased from Sigma.

### 2.4. Cell Lysates, Western Blots and Densitometry

Cells were washed in PBS (phosphate-buffered saline) and were lysed with RIPA buffer (50 mM Tris-HCl pH 8.0, 150 mM NaCl, 1% NP-40, 0.1% SDS, 0.5% sodium deoxycholate) for 10 min on ice. Lysates were cleared for 10 min at 13,000 rpm at 4 °C. Hypotonic lysates were prepared as previously reported [40].

Bradford assay, SDS-PAGE and Western blots were performed according to standard procedures (Bio-Rad system). Nitrocellulose membranes were incubated over night with indicated primary antibodies (working dilutions were 1:500 to 1:3000), secondary antibodies were incubated for 2 h at room temperature (working dilutions were 1:10,000 or 1:20,000). Blots were probed with GAPDH (glycerinaldehyde-3-phosphate-dehydrogenase) antibodies as normalization control. Protein signals were visualized and quantified within a linear range of exposure using a Li-COR ODYSSEY Imager and Image Studio Light software. Intermediate exposure times were chosen for data analysis, whereas overexposure (indicated by the imager) was excluded for measurements in accordance to previous studies [43]. For quantification, only properly blotted Western blots (lacking artificial spots that might influence the band intensities in false-positive or false-negative manner) were quantified. For normalization, protein band intensities were normalized to the respective GAPDH signals of the same lysate and blot. For preparation of figures, images were eventually processed using Adobe Photoshop CS6 software. 

### 2.5. Mitochondria Isolation and Proteinase K Treatment

Mitochondrial fractions were obtained using the Mitochondria/Cytosol fractionation Kit (BioVision) according to the manufacturer’s protocol and as shown previously [40]. Proteinase K and Triton X-100 (1%) were added on ice for 15 min and proteinase K (PK) was subsequently inactivated with PMSF (Sigma). 

### 2.6. Isolation of Recombinant Proteins and Lipid Binding Assay

Expression of GST or GST-PPAN constructs were induced by IPTG (1 mM) and purified from *E. coli* BL21(DE3) using glutathione sepharose (GE Healthcare). Proteins were analyzed by Coomassie staining and Western blotting. Membrane lipid strips (Echelon) were incubated with 1 µg/mL of each recombinant protein at 4 °C and detected by anti-GST antibodies according to the manufacturer’s instructions and as shown previously [44].

### 2.7. Immunofluorescence

Cells were grown on glass cover slips and were treated with drugs as indicated. Cells were fixed with ice-cold methanol for 10 min and permeabilized with 0.5% Triton X-100 for 15 min. Cells were blocked in 0.5% BSA for 45 min, incubated with primary (working dilutions were 1:50 to 1:150) and secondary antibodies (1:300 for Cy2 or Cy3, and 1:300 or 1:2000 for the Alexa 488 and 594 antibodies) for 1–2 h each and mounted in DAPI mounting medium. Secondary antibodies were tested for absence of background signals.

### 2.8. qPCR Expression Analysis

Total RNA from siRNA-transfected cells was purified with the RNeasy Mini Kit from Qiagen. After DNAse treatment (Roche), cDNA was transcribed with random hexamer primers and SuperScriptII (Invitrogen). In addition, samples without reverse transcriptase treatment served as contamination controls. Primers were designed using the PrimerQuestTool (IDT) and primer efficiency was calculated for HeLa cells with cDNA produced from control siRNA transfected cells as described [45]. Only primers that show efficiency between 90–110% (E = 1.8 to 2.2) were used. For qPCR analysis, the QuantiTect SYBR Green PCR Kit (Qiagen) was used on a BioRad CFX Connect Real Time System. Initially, the relative expression ratio R was calculated according to the efficiency corrected method for all HeLa data (E = 1.8 to 2.2) [45]. E = 2 was used for HEK293A GFP-LC3 and U2OS data. Each experiment was performed at least 4 times and measured in triplicates. Resulting *PPAN* qPCR products were verified by sequencing (eurofins) as well as by gel electrophoresis. Primers were purchased from biomers, the sequences were:PPAN-fw: 5-CGCCCAGGGAAGAGAGT-3PPAN-rv: 5-GAAACCGAGGGCCACATC-3GAPDH-fw: 5-TGGTCTCCTCTGACTTCAAC-3GAPDH-rv: 5-CGTTGTCATACCAGGAAATGAG-3

### 2.9. mtDNA Assay

Total DNA was purified with the DNeasy Blood and Tissue Kit (Qiagen). 1 ng DNA served as a template to determine the amount of mitochondrial and nuclear DNA content by qPCR, using specific primers for mt-tRNA**^Leu(UUR)^** and nuclear b2-microglobulin (B2M) as described [46]. The relative mtDNA was calculated as ratio of mt-tRNA**^Leu (UUR)^** to nuclear B2M (=mtDNA/nDNA ratio).

### 2.10. Image Acquisition and Cell Counting

If not otherwise indicated, confocal images were taken in a single plane with a Leica TCS SP5 II confocal microscope (63× objective) using LAS AF software and were eventually processed with Adobe Photoshop CS6 software. At least 200 cells per assay were counted on a Zeiss microscope (100× objective) in random optical fields and are presented as percentages related to the total cell number of the randomly selected optical fields. 

For LC3 puncta analysis, z-stack images of HEK293A GFP-LC3 cells (GFP fluorescence, co-stained with GAPDH (1:100)/Alexa 594 (1:1000) and DAPI) were taken with a Zeiss LSM710 confocal microscope (63× objective) using ZEN 2010B SP1 software (version 6.0). Slices in the z-plane were taken with an interval of 0.5 µm through the entire monolayer of cells. For each condition, 4 images in random optical fields were taken (approximately 50 cells per n). The respective z-stack images were subsequently processed with the z project module (ImageJ) using the Max intensity projection type, providing a composite image of all planes. Projected images were exported as TIFF, whereas the respective channels were retained. Quantitative analysis was executed with CellProfiler^TM^ (version 3.0.0) [47,48]. Nuclei of cells were defined as primary objects based on the DAPI staining. Afterwards, complete cells were defined as secondary objects by cytoplasmic GAPDH intensity with the respective nuclei as seeds. Finally, the number of GFP-LC3 puncta were analyzed and assigned to the particular parental cell. Only complete cells were involved in the quantification. In addition, the analysis of each image was manually reviewed. Mitotic and apoptotic cells (as indicated by DAPI counterstaining) were excluded from analysis as suggested earlier [17].

Slides of the same assay (e.g., si control and knockdown) were imaged and processed equally.

### 2.11. Transmission Electron Microscopy

HeLa cells were seeded on 4–6 carbon-coated sapphire discs per 6-well and were transfected as stated above. Cells on sapphire discs were high-pressure frozen, freeze substituted and in turn embedded in Epon (Fluka). Ultrathin sections were cut with an ultramicrotome (Leica) and sections were mounted on grids (Plano). The images were taken on a JEM1400 transmission electron microscope (Jeol).

### 2.12. Oxygraph Assay

Cellular oxygen uptake was quantified by high-resolution respirometry using the Oroboros Oxygraph-2K (Oroboros Instruments, Innsbruck, Austria). HeLa cells were transfected with siRNAs and trypsinized 48 h post transfection. Cell numbers were determined by trypan blue staining. Subsequently, the cells were resuspended in the two parallel chambers of the oxygraph calibrated for 2 mL of respiration medium at 37 °C, and continuously stirred at 750 rpm. Cellular respiration was quantified in terms of oxygen flux (*JO*_2_) based on the rate of change of the O_2_ concentration in the chambers after normalization to the total cell number: pmol O_2_/(s × 10^6^ cells). The initial *JO*_2_ obtained without exogenous addition of substrates or inhibitors represents the basal respiration rate of the intact cells. The experimental protocol was then started by injecting 1.25 µM Oligomycin to inhibit the ATP-synthase, thus yielding the LEAK-state. The *JO*_2_ in this state is determined by the respiratory rate required to compensate for proton leaking across the inner mitochondrial membrane, electron slipping along the respiratory chain and ion-exchange processes, and is an indicator of the coupling efficiency of the respiratory chain. 

The ATP-production related *JO*_2_ (*JO*_2_-ATP) is calculated as the difference between the basal respiration rate and the respiration rate at LEAK-state. Subsequently, the respiratory activity of the electron transport system (ETS) was measured in the uncoupled state obtained by continuous titration of the uncoupler p-trifluoromethoxy-carbonyl-cyanide phenylhydrazone (FCCP) into the respiration medium. The continuous titration was performed using a TIP-2K titration-injection micropump (Oroboros Instruments, Innsbruck, Austria). This device allowed to continuously inject a 1 mM FCCP solution at a rate of 1 nL/s. The continuous titration was manually stopped once the FCCP-induced increase in cell respiration had reached a maximum. At this point, the average FCCP concentration in the respiration medium was 0.49 ± 0.15 µM. Then the cells were permeabilized by adding 1.62 µM digitonin and maximum respiration was re-established by exogenous addition of the substrates malate (2 mM), glutamate (10 mM), and succinate (10 mM). This step was performed to test for maximum mitochondrial respiration under non-limiting substrate concentrations. Finally, after sequentially blocking complex I by 0.5 µM rotenone and complex III by 5 µM Antimycin A, selective complex IV activity was tested by addition of ascorbate (2 mM) and tetramethylphenylendiamine (TMPD, 0.5 mM), and corrected for the auto-oxidation of TMPD as previously described [49]. Control for the TIP-2K as well as data acquisition and analysis were performed with the DatLab software, version 7.0 (Oroboros Instruments). This software enables continuous monitoring and recording of the oxygen concentration in the chambers as well as of the derived oxygen flux over time, normalized for the amount of cells at rates of 0.5–1 Hz.

### 2.13. ATP, ADP and AMP Measurements

3000 HeLa cells were seeded per 96-well and were transfected with 83 nM siRNAs using Oligofectamine reagent as previously [40]. 

Each sample per independent experiment (n) was prepared and analyzed in biological triplicates, the mean values were used for statistical analysis. 48 h post-transfection, cells were used to measure ADP, ATP and AMP. Cells treated in parallel were counted with trypan blue staining. For normalization purposes, the total cell number was used to calculate the ratio per cell. 

The ATPlite Luminescence Assay kit from Perkin-Elmer was used to measure relative ATP. The ApoSENSOR ADP/ATP kit from EnzoLifeSciences was used to measure ADP/ATP ratio. The AMP/ATP ratio was measured using a combination of the AMP-Glo Kit from Promega and the ATPlite Luminescence Assay kit from Perkin-Elmer. Note that the kits were used according to the manufacturer’s instructions with one exception: Cells subjected to AMP measurements (AMP-Glo Kit from Promega) were initially lysed with the ATPlite lysis buffer from Perkin-Elmer according to the manufacturer’s instructions, to release nucleotides from the seeded, adherent cells. 

Positive controls (see Appendix A) were treated with staurosporine (STS) for 6 h (330 nM) prior to lysis to trigger apoptosis [40]. DMSO treated samples served as respective negative controls. 

17% to 20% of the lysates were used for luminescence measurements in 96-well format using the SpectraMax i3x multimode reader from Molecular Devices. 

### 2.14. Citrate Synthase Measurements

HeLa cells transfected in 6-well plates were collected 48 h post transfection. Measurements were performed as reported in the following protocol: http://wiki.oroboros.at/index.php/MiPNet17.04_CitrateSynthase. 

The measured values (Ultrospec 2100 pro, Amersham Biosciences) were normalized to the total cell number.

### 2.15. Statistical Analysis

Statistical analysis was performed with Excel and GrapPad Prism software version 6.00 using at least 3–4 independent experiments (n). If not otherwise indicated, the non-parametric Mann–Whitney rank sum test (one-tailed) was performed for non-normalized data sets. Analyses of single cells from HEK293A confocal experiments were calculated by unpaired t-test as indicated.

Statistical significance of relative data sets (e.g., Western blotting) was calculated with the paired t-test (one-tailed) and a normal distribution was assumed here as reported [43]. Normalization was performed for fixed si controls where indicated in the respective figure legends. Error bars indicate S.D., statistically significant differences are indicated by asterisks and are denoted in or at the bottom of the respective figure legends.

## 3. Results

### 3.1. PPAN Interacts with Cardiolipin and Controls Mitochondrial Homeostasis

We have previously shown that endogenous PPAN specifically localizes to mitochondria in HeLa cervical cancer cells, besides its well-known localization to nucleoli. In addition, we showed that siRNA-mediated knockdown of PPAN by two previously published and specific siRNAs, decreased nucleolar as well as mitochondrial PPAN pools [40]. 

Here, we first recapitulated the mitochondrial localization of endogenous PPAN in different cancer cell lines such as HCT116 colorectal cancer and U2OS osteosarcoma cells by co-localization with the mitochondrial inner membrane marker TIM23 (translocase of the inner membrane 23). In these cells, we detected PPAN also in the nucleoli and mitochondria as determined by confocal imaging (Figure 1A). 

We had earlier shown that a C-terminal PPAN-EGFP construct was sufficient for mitochondrial targeting and additionally identified a putative pre-sequence located in the N-terminus of PPAN, which might be required for mitochondrial import [40]. However, the precise sub-mitochondrial localization of PPAN remained unknown. 

Therefore, we here characterized the sub-mitochondrial localization of PPAN in more detail. To determine whether PPAN is localized at the mitochondrial outer membrane (MOM) or is imported and present at the mitochondrial inner membrane (MIM) or matrix, we isolated mitochondria from HeLa cells and performed a biochemical proteinase K (PK) assay in either presence or absence of Triton X-100 [50,51]. As controls we used antibodies against TOM20 (translocase of the outer membrane 20) for the MOM and TIM23 and OPA1 (optic atrophin 1) for the MIM. Our results demonstrated that the MOM protein TOM20 was efficiently degraded by PK treatment alone, whereas PPAN was partially resistant to PK treatment (Figure 1B). This finding indicated a localization of a fraction of PPAN at the MOM as well as another fraction inside mitochondria, such as inner surface of the MOM or MIM or matrix. To test this, we co-treated mitochondrial extracts with PK and Triton X-100, to allow perforation of mitochondrial membranes and access of PK to proteins of the MIM and mitochondrial matrix. This co-treatment led to a robust digestion of PPAN in a similar manner as of the MIM proteins OPA1 and TIM23 (Figure 1B). We conclude that a fraction of PPAN can be imported into mitochondria, which is presumably localized in the IMS and/or matrix. 

Next, we analyzed the amino acid composition of PPAN, given that cationic charge might be involved in mitochondrial targeting of PPAN. We found that basic amino acids (K,R,H) were clustered in the C-terminus of PPAN (Appendix A). We speculated that PPAN might interact with negatively charged mitochondrial membrane lipids such as cardiolipin (=CL) [52]. Although cardiolipin is mostly present at the MIM, it is also enriched at mitochondrial contact sides, which are micro domains formed between the MIM and MOM that can exchange by a flip-flop mechanism [53,54,55].

Using isolated recombinant GST (glutathione-S-transferase)-tagged PPAN we found a strong and specific interaction of GST-PPAN with cardiolipin (=CL) when performing in vitro lipid binding assays (Figure 1C–E). In contrast, GST itself did not bind to any of the tested cellular lipids (Figure 1C,D) as previously reported [44]. We also determined binding of the N- and C-terminal GST tagged PPAN deletion mutants (Figure 1C,D) to further map for a putative interaction domain. We thus incubated similar doses of each recombinant protein per lipid strip and found that both deletion mutants of PPAN bound to cardiolipin. However, the signal was weaker when compared to full-length GST-PPAN (Figure 1C,D), indicating that both domains might contribute to the interaction with cardiolipin. Unexpectedly, we found a strong interaction of C-terminal and full-length PPAN with sulfatides, which was not observed in the GST negative control (Figure 1C,D).

The striking sub-mitochondrial localization and interaction with cardiolipin led us to determine, whether the anti-apoptotic factor PPAN maintains mitochondrial integrity. For this purpose, we transiently depleted PPAN in HeLa cells and monitored the mitochondrial phenotype and function. 

Note that we previously reported a 50% decrease of PPAN upon siRNA transfection by independent siRNAs [40] (and Figure 2A), which might be considered low. However, depletion of key ribosome biogenesis factors is tightly linked to apoptosis induction. Additionally, ribosomopathies, a class of diseases associated with impaired ribosome biogenesis, are characterized by haploinsufficiency suggesting that null mutations are embryonic lethal [5]. Similarly, knockout of the ribosome biogenesis factor and PPAN binding partner, pescadillo (Pes1), has been shown to be lethal at the 2-cell stage in mice [56]. Therefore, we assume that PPAN knockdown cells with a higher knockdown efficiency might be cleared by apoptosis. We therefore decided to monitor approximately 50% of PPAN depletion. Throughout the manuscript, we investigated knockdown with the previously characterized PPAN siRNA termed si PPAN-B. To rule out off-target effects, key experiments were recapitulated with a second, in HeLa cells less efficient siRNA termed si PPAN-A (Figure 2A) [40].

To determine the functionality of mitochondria in PPAN knockdown cells, we measured mitochondrial respiration of intact and permeabilized HeLa cells. At this stage, we observed that mitochondrial respiratory function is affected in a specific manner by PPAN knockdown. In fact, we found only the ATP production-related respiratory activity of the HeLa cells to be reduced by PPAN knockdown (Figure 2B,C). In contrast, in the uncoupled state the respiratory activity of the PPAN knockdown cells remained unaffected (Figure 2B,D). This suggested that upon PPAN down-regulation the ATP turnover can already be reduced, albeit the available capacity of the respiratory chain is not yet the limiting factor. 

To determine relative ATP, ADP and AMP levels in PPAN knockdown cells, we performed luminescence-based measurements. PPAN knockdown in 96-well format was performed as previously [40] and was verified by Western blotting (Appendix A). Interestingly, we found that ATP levels in PPAN knockdown cells were significantly increased by two PPAN siRNAs (Figure 2E), whereas ADP/ATP and AMP/ATP ratios were not affected (Figure 2F,G). 

As positive control, we monitored effects of the pro-apoptotic drug staurosporine (STS). Western blotting against cleaved PARP revealed activation of caspases and induction of apoptosis upon STS treatment (Appendix A). STS treatment strongly reduced ATP levels under the tested conditions (Appendix A). In line, ADP/ATP and AMP/ATP ratios were strongly increased in STS treated samples, demonstrating functionality of the setup (Appendix A). 

Given the increase of ATP upon PPAN depletion, we measured citrate synthase activity as a gold standard for evaluating mitochondrial mass. We detected a significant increase of 24% in citrate synthase activity in si PPAN-B transfected HeLa cells, indicating increased mitochondrial mass (Figure 2H). We also evaluated COX IV (cytochrome *c* oxidase subunit IV), given that COX proteins are widely used as mitochondrial mass markers [17,57]. In contrast, we did not detect changes in COX IV by Western blotting upon PPAN knockdown (Appendix A). We then determined the relative mitochondrial DNA (mtDNA) to nuclear DNA (nDNA) ratio by qPCR as described [46], using the established mitochondrially-encoded tRNA**^Leu (UUR)^** and the nuclear target b2-microglobulin (B2M). We found no differences in mtDNA/nDNA ratio in HeLa cells transfected with both PPAN siRNAs (Appendix A). Overall, our data may suggest that measurements of citrate synthase activity might be more sensitive for detecting mitochondrial mass changes above a certain threshold.

Next, the mitochondrial phenotype of PPAN knockdown cells was monitored by transmission electron microscopy (TEM). PPAN depletion by two siRNAs resulted in strong defects of mitochondrial ultrastructure; and in particular, the MIM was disorganized when compared to controls (Figure 2I–K and Appendix A). Mitochondria of PPAN siRNA transfected cells displayed altered and perturbed cristae, whereas the MOM seemed largely unaffected (Figure 2I,K and Appendix A). Moreover, mitochondria were characterized by a considerably shorter and fragmented phenotype, presumably indicating induced mitochondrial fission. We also noticed appearance of swollen mitochondria. In summary, our data reveal that the PPAN knockdown phenotype is characterized by disrupted mitochondrial integrity and a dramatic disorganization of the inner membrane architecture. Moreover, the TEM data suggest an imbalance of mitochondrial fusion and fission following PPAN down-regulation.

### 3.2. PPAN Knockdown Enhances Basal Autophagy in HeLa Cells

Given the accumulation of defective mitochondria in HeLa cells, which are largely considered to be incapable of Parkin-mediated mitophagy [24], led us to determine whether PPAN depletion might affect basal autophagic flux as one alternative stress response mechanism.

Therefore, we knocked down PPAN and simultaneously blocked autophagic flux by the late stage autophagy inhibitor chloroquine (CQ) for 3.5 h to induce the accumulation of lipidated LC3-II as reported [58] (Figure 3A, right). The lipidated LC3-II can be discriminated from non-lipidated LC3-I by Western blotting. Although larger in mass, LC3-II shows a faster migration in the SDS gel when compared to LC3-I [17]. Indeed, we noticed that CQ-mediated LC3-II accumulation was significantly increased by over 50% following PPAN knockdown when compared to control transfected cells, reflecting activation of basal autophagic flux (Figure 3A–C). 

To provide further proof, we performed TEM imaging on control and si PPAN transfected HeLa cells following CQ treatment. We noticed an accumulation of autophagosomes in CQ positive control siRNA transfected cells treated for 3.5 h (Figure 3D and Appendix A) and 24 h with CQ (Figure 3F). Consistent with our immunoblot-based flux analysis, PPAN depletion by two siRNAs further increased number of cargo-filled autophagosomes/autolysosomes, which partially were also increased in size (Figure 3D–G and Appendix A). We noticed that more mitochondria were excluded from autophagosomes and autolysosomes in the si controls than in the PPAN knockdown cells (Figure 3D–G and Appendix A).

### 3.3. PPAN Knockdown Enhances Starvation-Induced Autophagic Flux in HeLa Cells

Next, we examined effects of PPAN knockdown in response to nutrient deprivation [17]. Transfected HeLa cells were incubated with EBSS medium (E, Earle’s balanced salt solution) to induce starvation-mediated autophagy in absence and presence of the autophagy inhibitor Bafilomycin A1 (B, BafA1) for 3.5 h. Again, PPAN knockdown cells showed higher basal autophagic flux in presence of the autophagy inhibitor BafA1 (Figure 4A,B). In the positive control, BafA1 and EBSS co-treatment increased LC3-II levels further when compared to BafA1-treated controls (Figure 4A,B). The highest levels of LC3-II were obtained by BafA1 and EBSS co-treatment in the PPAN knockdown cells, thereby demonstrating that PPAN depletion enhances autophagic flux also in response to nutrient deprivation. Given the rather weak signals of LC3-II by Western blotting in EBSS or BafA1 treated control samples in these settings, we verified the functionality of these treatments. To this end, we measured turnover of the autophagy receptor p62 (SQSTM1, sequestosome1; hereafter referred to as p62), which contains an LC3 interaction motif (LIR) as well as a ubiquitin associated domain for binding to LC3 and ubiquitin, respectively. As a result, autophagosomal membranes are selectively recruited to ubiquitinated cargo by p62 [21]. EBSS-mediated autophagy induction revealed increased degradation of the autophagy substrate p62, whereas BafA1-mediated inhibition led to significant increase of p62 (Appendix A). 

### 3.4. PPAN Knockdown Enhances Basal Autophagic Flux in HEK293A GFP-LC3 Cells

To examine the effect of PPAN knockdown on autophagic flux in another cell line, we used stable HEK293A GFP-LC3 cells, which are widely used for flux studies [17]. We verified knockdown of PPAN by both siRNAs on protein (Figure 5A,C) and/or mRNA level (Appendix A). Similar as in HeLa cells, we detected 50% decrease of PPAN protein (Figure 5A,C and Appendix A) or 40% decrease of *PPAN* mRNA by siPPAN-A (Appendix A). 

We then analyzed the autophagy activity in response to CQ treatment in control and PPAN knockdown HEK293A GFP-LC3 cells (Figure 5A,B and Appendix A). We quantified levels of lipidated GFP-LC3-II. Upon CQ treatment, we detected 25% or 50% increase of GFP-LC3-II after PPAN knockdown by both PPAN siRNAs (Figure 5A,B and Appendix A). 

When performing immunofluorescence-based microscopy in HEK293A GFP-LC3, we found that PPAN depletion in absence of CQ led to accumulation of cells with GFP-LC3 puncta, representing autophagosomes (Figure 5D and Appendix A). In CQ treated controls, almost 100% of the cells were positive for massive puncta accumulation, in line with efficient block of autophagy (Figure 5D and Appendix A). Consistent with enhancement of autophagy, we found a significantly increased number of GFP-LC3 puncta per cell in presence of CQ in the PPAN knockdown cells compared to controls (Figure 5D–F and Appendix A). We also noticed that more punctae of larger size accumulated the in PPAN knockdown than in the CQ controls (Figure 5E,F and Appendix A).

### 3.5. PPAN Knockdown Leads to Accumulation of the Mitophagy Machinery at Mitochondria in Parkin-Expressing HeLa Cells

The accumulation of damaged mitochondria led us investigate if PPAN knockdown stimulates mitophagy. Parkin-dependent mitophagy requires the action of the ubiquitin ligase Parkin, essential for ubiquitination of mitochondrial cargo [25]. Being aware that the cellular HeLa system expresses almost no endogenous Parkin [59,60] and thus is considered to be incompetent for Parkin-dependent mitophagy [24], we investigated the situation in a Parkin positive background. To do so, we transiently co-transfected EGFP-Parkin and control or PPAN siRNAs in HeLa cells and treated them with the mitochondrial protonophore uncoupler CCCP to trigger mitophagy [61]. CCCP represents one of the first drugs used to study mitophagy by inducing mitochondrial depolarization [13,24,62]. CCCP induces mitochondrial condensation in peri-nuclear regions, Parkin recruitment to mitochondria, which is followed by Parkin-mediated mitophagy as CCCP treatment proceeds. However, CCCP is also known to stimulate (macro)autophagy by inducing LC3 lipidation [63]. 

We then monitored mitochondria by HSP60 staining in this so-called “enhanced mitophagy setup” upon Parkin overexpression [24]. Whereas the control siRNA transfected cells performed mitophagy only when stimulated by CCCP, we noticed a significant induction of mitochondrial condensation and EGFP-Parkin recruitment to mitochondria in 25% of the DMSO controls following PPAN knockdown, which was in the same range as the CCCP positive controls. PPAN knockdown cells showed significantly more cells with EGFP-Parkin co-localizing with mitochondrial HSP60 (Figure 6(A,B,C1)). Moreover, we noticed co-localization of residual mitochondrially-recruited EGFP-Parkin with ubiquitin, which represents earmarks for mitophagic degradation in PPAN knockdown cells (Figure 6(C2,D)). We also found EGFP-Parkin co-localization with endogenous LC3 following PPAN knockdown (Figure 6(C3,E)), pointing to engulfment of ubiquitinated, mitochondrial cargo into LC3 autophagosomes. In addition, we detected accumulation of LAMP2 (lysosome associated membrane glycoprotein 2)-positive lysosomes surrounding recruited EGFP-Parkin, indicating recruitment of lysosomes to mitochondria following PPAN knockdown (Figure 6(C4,F)). Taken together, our data indicate that PPAN knockdown induces mitochondrial damage, which in turn triggers Parkin recruitment to mitochondria followed by autophagosomal engagement and lysosomal delivery of damaged mitochondria.

### 3.6. PPAN is Decreased by CCCP and PPAN Knockdown Further Decreases Mitochondrial Mass in Parkin-Expressing Cells

The effects of PPAN knockdown observed in EGFP-Parkin overexpressing HeLa cells led to the assumption that PPAN loss might promote mitophagy in Parkin-expressing cells. 

Therefore, we next investigated the situation in HEK293 and HCT116 cells expressing abundant endogenous Parkin levels [24,60,64]. Firstly, we found that endogenous PPAN also localizes in nucleoli and mitochondria in these cells (Figure 7A and compare Figure 1A). We then determined COX IV protein in whole cell lysates by Western blotting in absence or presence of CCCP.

In absence of CCCP we found that the mitochondrial mass marker COX IV was significantly decreased to 80% following PPAN depletion in HEK293A GFP-LC3 cells (Figure 7B–D). CCCP also decreased COX IV to 46% in controls (Figure 7B,C). PPAN knockdown cells treated with CCCP had significantly lower COX IV levels of 38% than the CCCP-treated control cells (Figure 7B,C). This indicates that PPAN depletion is associated with loss of mitochondrial mass and induction of mitophagy in these cells. When stimulating control siRNA transfected cells with CCCP, we noticed a decrease of PPAN by 29% (Figure 7D). 

In HCT116 cells the same was true (Figure 7E–G). Additionally, here, PPAN knockdown was associated with significantly decreased COX IV levels of 79% (Figure 7E,F). CCCP treatment of controls decreased COX IV to 46%. PPAN was also significantly decreased upon CCCP treatment to 78% in control siRNA transfected cells (Figure 7E,G). In addition, CCCP treatment led to a significant 3-fold increase of LC3-II in controls, which was further increased in PPAN knockdown cells to 9-fold (Figure 7E,H). 

Taken together, our data show that PPAN depletion sensitizes cells for mitophagy as it further promotes decrease of COX IV. PPAN itself is decreased by CCCP in Parkin-expressing cells, being in line with the fact that PPAN localizes to mitochondria and thus might itself be subjected to autophagosomal degradation together with mitochondria. 

### 3.7. PPAN Knockdown Decreases Mitochondrial Mass and mtDNA to Nuclear DNA Ratio in U2OS Cells

We next determined mitophagy in U2OS cells. We made use of the physiologically relevant mitophagy inducers Oligomycin and Antimycin A (O/A) [62] and monitored induction of mitophagy in this cell line. O/A treatment inhibits the respiratory complex and ATP synthase and triggers a more specific and milder response than CCCP [62]. After 24 h of O/A treatment, COX IV protein levels were decreased to 44% indicating induction of mitophagy in U2OS cells (Figure 8A,B). At the same time, PPAN levels were not significantly altered after 24 h of O/A treatment (Figure 8A,C). Knockdown of PPAN in U2OS cells led to a significant decrease of COX IV protein by Western blotting (Figure 8D–F), indicating decrease of mitochondrial mass.

To quantify the mitochondrial DNA ratio in U2OS cells upon PPAN knockdown, we isolated total DNA from control and PPAN siRNA transfected cells. We confirmed PPAN knockdown on mRNA level and detected a significant down-regulation of *PPAN* by 53% for si PPAN-B and 63% for si PPAN-A in U2OS cells (Figure 8G). We then determined the mtDNA/nDNA ratio by qPCR. We detected a significant decrease of 42% and 24% of the mtDNA ratio upon PPAN knockdown, when using two PPAN siRNAs (Figure 8H). Likewise, O/A treatment for 24 h resulted in 37% decrease of mtDNA/nDNA ratio in U2OS cells (Figure 8I).

Together, this shows that PPAN depletion is associated with a decrease of mitochondrial mass and mitochondrial DNA content by independent methods in various Parkin-expressing cell lines.

## 4. Discussion

Nucleolar stress is tightly coupled to mitochondrial impairment, thereby mimicking early hallmarks of neurodegenerative diseases [65]. However, the mechanisms that link both branches are still far from being well understood. 

We herein demonstrate for the first time that PPAN knockdown, which was earlier linked to nucleolar stress induction, promotes autophagy and mitophagy. In addition, we uncover that PPAN sustains mitochondrial integrity, whereas PPAN knockdown impairs mitochondrial function. We assume that the mitochondrial pool of PPAN directly coordinates mitochondrial homeostasis. It seems likely that PPAN depletion, by impairing ribosome biogenesis, might abrogate mitochondrial maturation and/or biogenesis. However, these aspects still have to be uncovered in the future. 

Here, we have used different cellular models, experimental setups and PPAN knockdown by two independent PPAN siRNAs to decrease the likelihood of off-target effects. Note that we could not yet perform rescue experiments with PPAN owing the localization of overexpressed full-length PPAN to nucleoli, but not to mitochondria [40]. In addition, simultaneous PPAN knockdown and transient PPAN expression was toxic for cells (not shown). 

To elucidate the mechanisms underlying the mitochondrial function of anti-apoptotic PPAN, we determined its sub-mitochondrial localization. Our analyses on PPAN reveal a direct interaction with cardiolipin and a possible import of a fraction of PPAN into mitochondria. In support of this, proteins can be translocated to the matrix via cardiolipin interaction [66]. Cardiolipin binding proteins were shown to play a crucial role for mitochondrial function and cristae morphology [67], all of which are affected following PPAN knockdown. Therefore, we propose that PPAN functions in an anti-apoptotic manner by sustaining mitochondrial surveillance. We speculate that PPAN might counteract BAX-dependent mitochondrial outer membrane permeabilization [40] by its interaction with cardiolipin, presumably also at mitochondrial contact sides. Cardiolipin mediates import of cytosolic proteins through these contact sides. As a matter of fact, cardiolipin located at the MOM serves as docking site for pro-apoptotic tBid, thereby facilitating BAX-dependent mitochondrial outer membrane permeabilization [33,34,68,69]. Cardiolipin functions further as an essential platform for translocation of caspase 8 to the outer membrane and its activation [70,71]. Cardiolipin is essential for cristae remodeling and subsequent release of cytochrome *c* [68,72]. Importantly, impaired cardiolipin biosynthesis, as observed in patients with Barth syndrome, is linked to altered cristae structures [73]. Although we have no direct evidence for a cardiolipin-dependency, PPAN knockdown at least recapitulates these effects. 

When considering the striking impact of PPAN knockdown on mitochondrial morphology, and in particular the altered cristae observed by TEM, it seems reasonable to expect corresponding impairment of the mitochondrial respiratory function. Surprisingly, mitochondrial respiration is less severely affected by PPAN knockdown. Although ATP-related oxygen consumption is significantly reduced following PPAN knockdown, ATP levels in PPAN knockdown cells are increased. Therefore, oxidative phosphorylation might be an efficient source of energy in PPAN knockdown cells. Together with the finding of increased citrate synthase activity, we assume that PPAN knockdown cells might display several compensatory mechanisms that keep ATP levels high: decrease of their metabolic rate, increase of mitochondrial mass, or efficient oxidative phosphorylation. Our data may also indicate that, as a consequence of a cardiolipin-PPAN interaction, modulatory effects on the ATP-turnover of the mitochondria might be mediated by PPAN. In fact, cardiolipin is known to strongly interact at the molecular level with both the adenine nucleotide translocase and the ATP-synthase, thus suggesting a potential role on the regulation of ATP-turnover [74,75,76,77,78]. However, this might not be the limiting factor of HeLa cells depleted of PPAN.

In this study, we further show by various established setups [17,79] that PPAN knockdown enhances basal autophagic flux. Autophagy is also stimulated in response to nutrient deprivation in PPAN knockdown cells. We clearly identified accumulation of autophagosomes following CQ- and BafA1-mediated lysosome inhibition, which are further increased following PPAN knockdown. In TEM flux studies we detected engulfment of various cellular cargo in autophagosomes and/or autolysosomes, being indicative for bulk autophagy. Therefore, a loss of the ribosome biogenesis factor PPAN engages induction of autophagy. 

In support of this notion, the nucleolus has recently been noticed as a central hub in the autophagic response [11]. Nucleolar stress, by interfering with RNA Polymerase I function, was suggested as upstream trigger of autophagy [9,10,80]. The pro-autophagy response following PPAN depletion might therefore be a consequence of nucleolar stress induction. 

HeLa cells used in this study are considered to be incompetent to perform Parkin-dependent mitophagy [24], given a lack of Parkin expression [59,60]. However, other mitophagy mechanisms exist as well [81,82,83]. In cell lines (over-)expressing (EGFP-)Parkin, we demonstrate an involvement of mitophagy upon PPAN depletion. In favor of our assumption we observed mitochondrial condensation around peri-nuclear regions, recruitment of EGFP-Parkin to ubiquitinated mitochondria and decrease of mitochondrial mass markers in Parkin-expressing cells. This finding is in accordance to studies showing that mitochondrial depolarization or ubiquitination are well-accepted stimuli for mitophagy induction. In support of this, we detected co-localization of EGFP-Parkin with ubiquitin at mitochondria and with LC3-positive autophagosomes. Additionally, EGFP-Parkin was found in close proximity to LAMP2-positive lysosomes, reinforcing this fact. This is reminiscent of CCCP- or O/A-mediated mitophagy of control siRNA transfected cells. We found that PPAN itself is decreased by CCCP-mediated mitophagy and that a loss of PPAN sensitizes cells for mitophagy.

It has been demonstrated that autophagosomal LC3 binds to cardiolipin, which is externalized from the MIM to the MOM following mitochondrial damage in order to engage mitophagy [84,85]. It was speculated that a factor spanning the IMS might be involved and that further cardiolipin interactors might restrict the redistribution of cardiolipin to the MOM [84,86]. Given the PPAN-cardiolipin interaction and similarities to our study we tempt to speculate that a loss of PPAN might trigger mitophagy by promoting access to cardiolipin. 

Note that strong mitochondrial stress triggers apoptosis [29]. How the transition from autophagy to apoptosis is precisely regulated is still not fully understood [28,87]. We suppose that in case of PPAN knockdown, cells cannot cope with the defective mitochondrial load solely by autophagy mechanisms. Likewise, we did not observe complete elimination of mitochondrial markers in the HeLa EGFP-Parkin system following PPAN knockdown. Therefore, it might well be that mitophagy is inhibited at some point. Consequently, the traffic jam might result in apoptosis induction as shown by us earlier [40]. Similar dual roles exist for Parkin, which dependent on the degree of the mitochondrial damage, sensitizes either towards mitophagy or apoptosis [88]. Given that autophagy requires a threshold of ATP [89], and also recovers ATP, high ATP as observed in PPAN knockdown HeLa cells, might initially facilitate autophagy and prevent apoptosis [90,91].

We speculate that PPAN loss—by triggering nucleolar stress and mitochondrial dysfunction—signals to autophagy and apoptosis. As a basis for this, we hypothesize that PPAN might stimulate a cellular self-protection mechanism upon alterations of nucleolar and mitochondrial integrity. Under basal levels of stress, PPAN loss is connected to initial protective autophagy mechanisms. In contrast, severe or continuous stress might drive the cells into apoptosis as a point of no return. In line with this, we have recently shown that pronounced depletion of PPAN increases the pro-apoptotic response in a time-dependent manner [41]. In favor of such a mechanism, a mouse model for nucleolar stress displayed a Huntington’s disease-like phenotype with autophagy induction as initial neuroprotective and life-extending mechanism, preceding apoptosis [8]. 

Overall, our study uncovers that loss of the ribosome biogenesis factor PPAN causes an imbalance of autophagic pathways and apoptosis. Given the emerging role of nucleolar stress and autophagy in aging, neurodegeneration and treatment of cancer [11], delineating the underlying molecular mechanisms will be beneficial for our understanding of these pathologies. 

## Figures and Tables

**Figure 1 cells-08-00894-f001:**
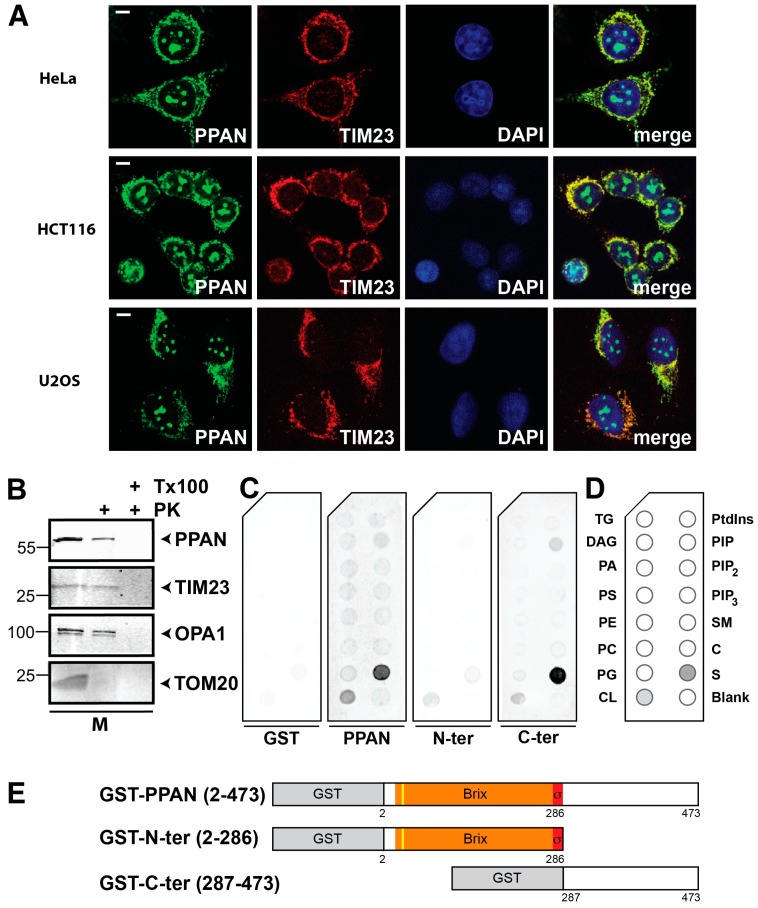
PPAN (Peter Pan) is localized within mitochondria and interacts with cardiolipin. (**A**) Endogenous PPAN localizes to nucleoli and mitochondria in human cancer cell lines as indicated. Mitochondria were co-stained with TIM23 antibodies. The merge shows co-localization of PPAN and TIM23. The images are representative of 3 (HeLa) or 4 experiments. Scale bar, 10 µm. (**B**) PPAN is localized at the MOM and within mitochondria, presumably at the mitochondrial inner membrane and/or matrix. Mitochondrial extracts (=M) from HeLa cells treated with proteinase K (=PK) and Triton X-100 (=Tx100) as indicated were subjected to Western blotting. Membranes were probed with antibodies against PPAN, mitochondrial inner membrane markers TIM23 and OPA1, as well as the mitochondrial outer membrane marker TOM20. Numbers indicate kDa. The Western blot is representative of 3 independent experiments. (**C**) PPAN interacts with the mitochondrial lipid cardiolipin as shown by cellular membrane lipid binding assays. Lipid strips were incubated with recombinant GST (glutathione-S-transferase) as negative control (left), or with either GST-PPAN or the GST-tagged PPAN deletion mutants N-ter and C-ter (right) as depicted. Lipid strips were simultaneously detected with an anti-GST antibody. Representative strips are shown. The assay was performed twice. Note that a schematic of the lipid strip is shown in (**D**) and the PPAN constructs are depicted in (**E**). (**D**) Schematic of the lipid strip used in (**C**). The internal blank is spotted in the lower right corner, the interacting lipids from (**C**) are highlighted in grey. TG, triglyceride; DAG, diacylglycerol; PA, phosphatidic acid; PS/PE/PC/PG, phosphatidyl-serine/-ethanolamine/-choline/-glycerol; CL, cardiolipin; Ptdins, phosphatidylinositol; PIP, phosphatidylinositol(4)phosphate; PIP_2_, Ptdins(4,5)phosphate; PIP_3_, Ptdins(3,4,5)phosphate; SM, sphingomyelin; C, cholesterol; S, sulfatide. (**E**) Schematic representation of GST-PPAN and the GST-tagged PPAN deletion mutants used in (**C**). GST-tag (grey), Brix domain (orange), σ70 motif (red), putative cleavable mitochondrial pre-sequence (yellow), as indicated [40]. Numbers indicate amino acid positions.

**Figure 2 cells-08-00894-f002:**
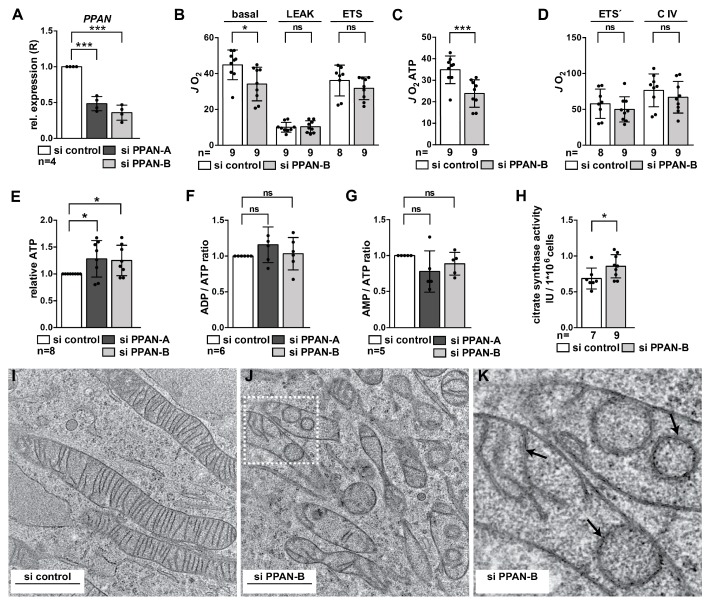
PPAN controls mitochondrial homeostasis. (**A**) qPCR analysis of HeLa cells transfected with si control and PPAN siRNAs as indicated. Transfection of both PPAN siRNAs results in significant down-regulation of *PPAN* mRNA levels. *PPAN* was normalized to *GAPDH* (=relative expression, R). Error bars represent S.D., *p* values were calculated by t-test. Statistically significant differences are indicated by asterisks, ***, *p* < 0.001; n = number of independent experiments. (**B**–**D**) Mitochondrial respiratory activity of intact, transfected HeLa cells under basal conditions, after inhibition of the ATP-synthase by Oligomycin (=LEAK-state), and under FCCP-induced respiratory uncoupling (=ETS-state). Only in the basal conditions, but not in the LEAK-state and/or ETS-state, mitochondrial respiration was reduced by PPAN knockdown (**B**). Accordingly, the ATP production-related oxygen flux (=*JO*_2_ATP) was significantly lower in the si PPAN-B transfected cells when compared to controls (**C**). After permeabilization of the outer cell membrane by digitonin and addition of exogenous substrates, the maximum respiratory activity (=ETS’) as well as selective complex IV activity (=C IV) were not significantly different between PPAN knockdown cells and si controls (**D**). Note that *JO2* was normalized to the total cell number: pmol O_2_/(s × 10^6^ cells). Error bars represent S.D.; *p* values were calculated by non-parametric Mann–Whitney rank sum test. Statistically significant differences are indicated by asterisks, *, *p* < 0.05, ***, *p* ≤ 0.001, ns, non-significant differences; n = number of independent experiments. (**E**–**G**) PPAN knockdown increases relative ATP levels in HeLa cells, whereas ADP/ATP and AMP/ATP ratios are not affected. HeLa cells were seeded in 96-well plates and were transfected with 83 nM siRNAs for 48 h as indicated. ATP levels normalized to the cell count, the ADP/ATP ratio and AMP/ATP ratio were detected by luminescence measurements; fold changes are depicted. Error bars represent S.D.; *p* values were calculated by t-test. Statistically significant differences are indicated by asterisks, *, *p* < 0.05, ns, non-significant differences; n = number of independent experiments. (**H**) Citrate synthase measurements of HeLa cells transfected with control and PPAN siRNAs as indicated. IU, international units. Error bars represent S.D.; *p* values were calculated by non-parametric Mann-Whitney rank sum test. Statistically significant differences are indicated by asterisks, *, *p* < 0.05; n = number of independent experiments. (**I**) HeLa cells were transiently transfected with control siRNA and cellular sections were analyzed for their mitochondrial phenotype by transmission electron microscopy (TEM). The assay is representative of 4 independent experiments. Scale bar, 1 µm. (**J**) HeLa cells were transfected with si PPAN-B. Representative cellular sections were imaged by TEM. Smaller mitochondria displaying impaired mitochondrial integrity and disrupted inner mitochondrial membranes following PPAN knockdown are shown. A representative image is shown, the TEM assay was performed 4 times. Scale bar, 1 µm. The boxed region is magnified in (**K**). (**K**) Magnification of the boxed area depicted in (**J**). The arrows point to perturbed cristae, indicated by membranous character.

**Figure 3 cells-08-00894-f003:**
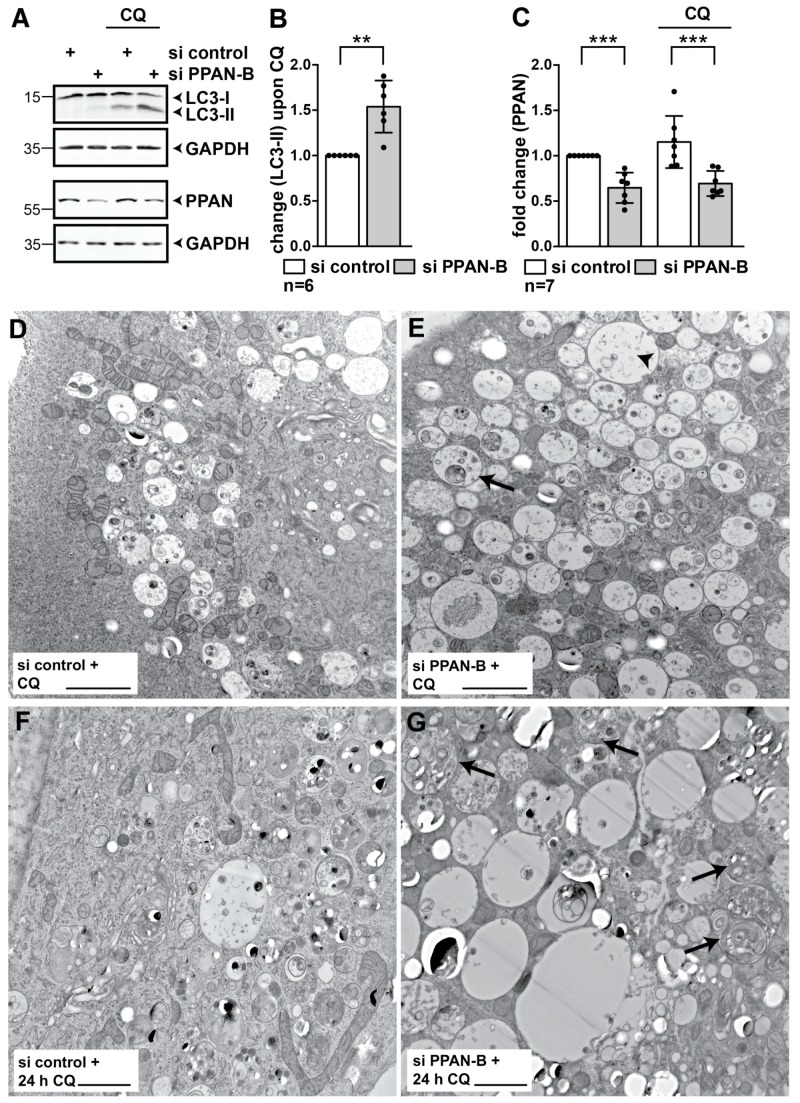
PPAN knockdown enhances basal autophagic flux in HeLa cells. (**A**–**C**) PPAN knockdown increases basal autophagic flux upon chloroquine treatment. (**A**) HeLa cells were transfected with siRNAs and were treated with the autophagy inhibitor chloroquine (=CQ) for 3.5 h as indicated above panels. PPAN knockdown results in significantly increased LC3-II levels following CQ treatment when compared to controls. Whole cell lysates were subjected to Western blotting and membranes were probed with total LC3, PPAN and GAPDH antibodies. The total LC3 antibody recognizes non-lipidated LC3-I and lipidated LC3-II. Whole cell lysates of the same experiment were loaded twice (see GAPDH placed below respective blots). Numbers indicate kDa. The LC3-II quantification is depicted in (**B**) and PPAN in (**C**). (**B**,**C**) Quantification of LC3-II normalized to GAPDH in presence of CQ (B) by densitometry as shown in (**A**). Quantification of PPAN normalized to GAPDH in presence or absence of CQ (**C**). The si control was set to 1. Error bars represent S.D., *p* values were calculated by t-test. Statistically significant differences are indicated by asterisks, **, *p* < 0.01, ***, *p* < 0.001; n = number of independent experiments. (**D**,**E**) PPAN knockdown enhances autophagosome accumulation in flux experiments as observed by transmission electron microscopy (TEM). (**D**) HeLa cells were transfected with control siRNA or si PPAN-B (**E**) and incubated with CQ for 3.5 h to block autophagy. Representative cell sections of two independent experiments are depicted. Scale bar, 2 µm. The arrow indicates accumulation of cargo-filled autophagosomes/-lysosomes, the arrowhead indicates expanded vesicles. (**F**,**G**) PPAN knockdown enhances autophagosome accumulation in flux experiments as observed by TEM. HeLa cells were transfected with control siRNA (**F**) and si PPAN-B (**G**) and incubated at 24 h post transfection for another 24 h with CQ. Representative cell sections of two independent experiments are depicted. Scale bar, 2 µm. The arrows indicate accumulation of cargo filled autophagosomes/-lysosomes.

**Figure 4 cells-08-00894-f004:**
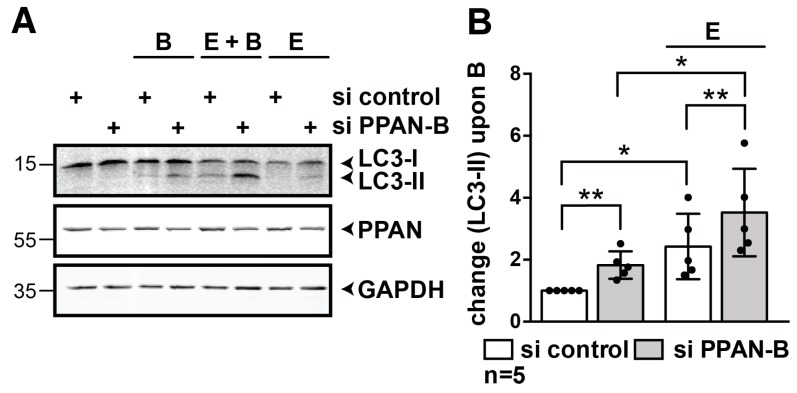
PPAN knockdown enhances autophagic flux in response to nutrient deprivation in HeLa cells. (**A**,**B**) PPAN knockdown increases basal autophagic flux in HeLa cells as shown by BafA1 treatment and promotes starvation-induced autophagy. Depletion of PPAN by siRNA significantly increases levels of LC3-II in presence of the lysosome inhibitor BafA1 for 3.5 h (=B). Following starvation-induced autophagy by EBSS (=E) for 3.5 h, PPAN knockdown further increases LC3-II in presence of BafA1 (=E + B). Western blotting and quantification was performed as in Figure 3, LC3-II normalized to GAPDH is shown in (**B**). The si control in presence of BafA1 was set to 1, and LC3-II was measured as fold change related to BafA1 in controls. Error bars represent S.D., *p* values were calculated by t-test. Statistically significant differences are indicated by asterisks, *, *p* < 0.05, **, *p* < 0.01; n = number of independent experiments.

**Figure 5 cells-08-00894-f005:**
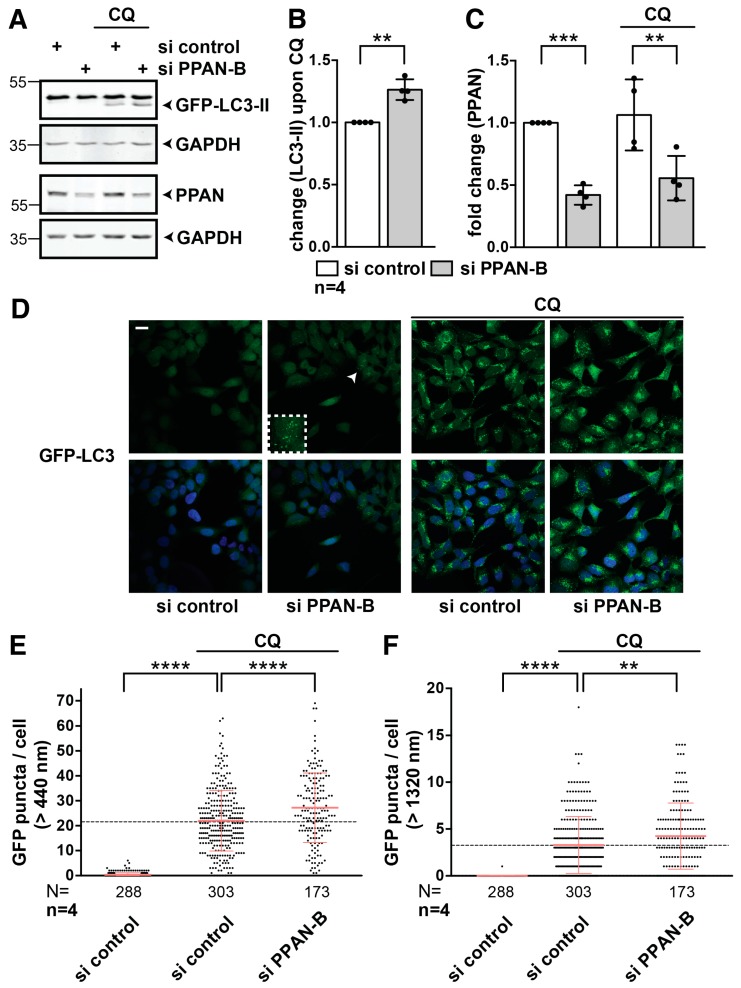
PPAN knockdown enhances basal autophagic flux in HEK293A GFP-LC3 cells. (**A**–**C**) GFP-LC3-II levels increase after PPAN knockdown in HEK293A GFP-LC3 cells in presence of CQ for 3.5 h. Western blots of whole cell lysates. Membranes were probed with LC3, PPAN and GAPDH antibodies, the statistical evaluation of LC3-II is shown in (**B**) and of PPAN in (**C**). (**B**) Quantification of LC3-II in presence of CQ normalized to GAPDH by densitometry as shown in (A) and of PPAN (**C**) in presence and absence of CQ as indicated. The si control was set to 1 as indicated. Error bars represent S.D., *p* values were calculated by t-test. Statistically significant differences are indicated by asterisks, **, *p* < 0.01, ***, *p* < 0.001; n = number of independent experiments. (**D**) PPAN knockdown by si PPAN-B increases accumulation of GFP-LC3 puncta in HEK293A GFP-LC3 cells. Representative confocal image stacks of randomly chosen optical fields are depicted, the upper panels show the GFP fluorescence, and the lower panels the merge with DAPI. For flux experiments, cells were treated with CQ for 3.5 h. Scale bar, 2 µm. The arrowhead points to representative GFP-LC3 puncta in PPAN knockdown samples, which are magnified in the boxed inset. (**E**,**F**) Random confocal image stacks of (**D**) were counted for GFP-LC3 positive puncta per cell, indicating autophagosome accumulation. The mean of all counted cells transfected and treated as indicated is depicted, in (**E**) all puncta larger than 440 nm, and in (**F**) puncta larger than 1320 nm were included for analysis. Error bars represent S.D., *p* values were calculated by unpaired t-test. Statistically significant differences are indicated by asterisks, **, *p* < 0.01, ****, *p* < 0.0001; n = number of independent experiments included in the analysis, N = total number of independently counted cells.

**Figure 6 cells-08-00894-f006:**
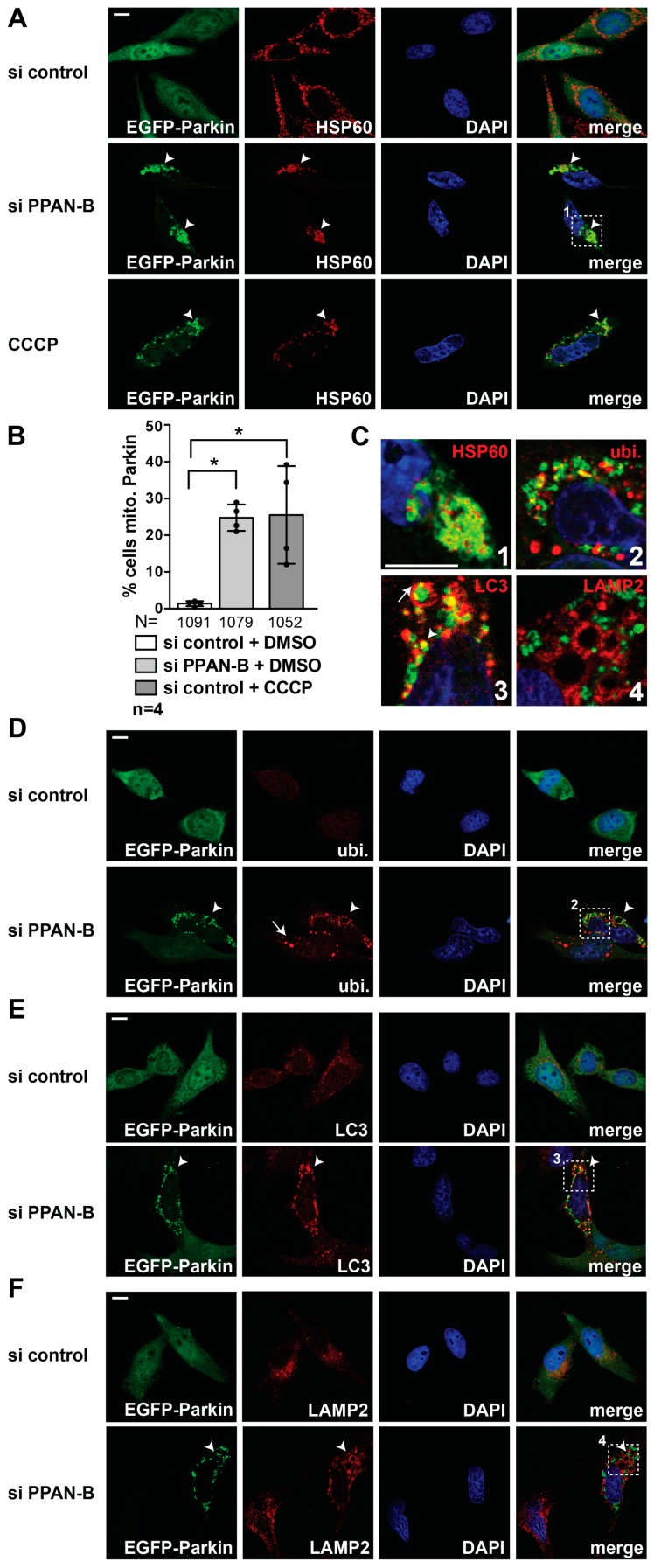
EGFP-Parkin and the mitophagy machinery are recruited to mitochondria in PPAN knockdown cells. (**A**,**B**) PPAN knockdown leads to mitochondrial recruitment of EGFP-Parkin and is accompanied by mitophagy induction in EGFP-Parkin overexpressing HeLa cells. (**A**) HeLa cells were co-transfected with EGFP-Parkin and either control siRNA or si PPAN-B as indicated. After 24 h of transfection, cells were treated with DMSO or the mitophagy inducer CCCP (1 µM) for 24 h. Cells were stained with HSP60 antibodies, nuclei with DAPI and EGFP-Parkin was detected by fluorescence. Arrowheads point to mitochondrially recruited EGFP-Parkin and residual HSP60 signal in cells. The boxed inset (1) is magnified in (**C**). Scale bar, 10 µm. (B) Quantification of EGFP-Parkin-positive cells recruited to mitochondria as shown in (**A**). Error bars represent S.D.; *, *p* < 0.05. n = number of independent experiments, N = number of randomly counted cells. (**C**) Merged insets magnified from si PPAN-B and DMSO treated samples shown in A (inset 1), D (inset 2), E (inset 3), F (inset 4) displaying mitophagy induction. (1) depicts HSP60 (red) co-localization with EGFP-Parkin (green) following PPAN knockdown. (2) shows ubiquitin (red) and mitochondrially recruited EGFP-Parkin (green) co-localization, (3) demonstrates co-localization of LC3-positive autophagosomes (red) with EGFP-Parkin (green) (arrowhead) and engulfment of mitochondrially recruited EGFP-Parkin into LC3 autophagosomes (arrow). (4) shows enlargement of LAMP2-positive lysosomes (red) surrounding recruited EGFP-Parkin (green). Representative merge images are depicted, nuclei were stained with DAPI. Scale bar, 10 µm. (**D**) PPAN knockdown triggers recruitment of EGFP-Parkin to ubiquitinated mitochondria. HeLa cells were co-transfected as in (**A**) and treated with DMSO. Cells were stained with ubiquitin (=ubi.) antibodies and DAPI, EGFP-Parkin was detected by fluorescence. Arrowheads point to mitochondrially recruited EGFP-Parkin and ubiquitin co-staining, the arrow demonstrates ubiquitination in EGFP-Parkin-negative or low-expressing cells. The boxed inset (2) is magnified in (**C**). A representative image is shown, the assay was performed 3 times. Scale bar, 10 µm. (**E**) PPAN knockdown induces EGFP-Parkin and LC3 co-localization, demonstrating engulfment of mitochondria by autophagosomes. HeLa cells were co-transfected as in (**A**) and treated with DMSO. Cells were stained with LC3 antibodies, nuclei by DAPI and EGFP-Parkin was detected by fluorescence. The arrowhead points to mitochondrially recruited EGFP-Parkin and LC3 autophagosomal co-staining. The boxed inset (3) is magnified in (**C**). A representative image is shown, the assay was performed 3 times. Scale bar, 10 µm. (**F**) PPAN knockdown triggers EGFP-Parkin accumulation in proximity to LAMP2-positive lysosomes. HeLa cells were co-transfected as in (**A**) and treated with DMSO. Cells were stained with LAMP2 antibodies, nuclei by DAPI and EGFP-Parkin was detected by fluorescence. The arrowhead points to mitochondrially recruited EGFP-Parkin in close proximity to enlarged lysosomes. The boxed inset (4) is magnified in (**C**). A representative image is shown, the assay was performed 3 times. Scale bar, 10 µm.

**Figure 7 cells-08-00894-f007:**
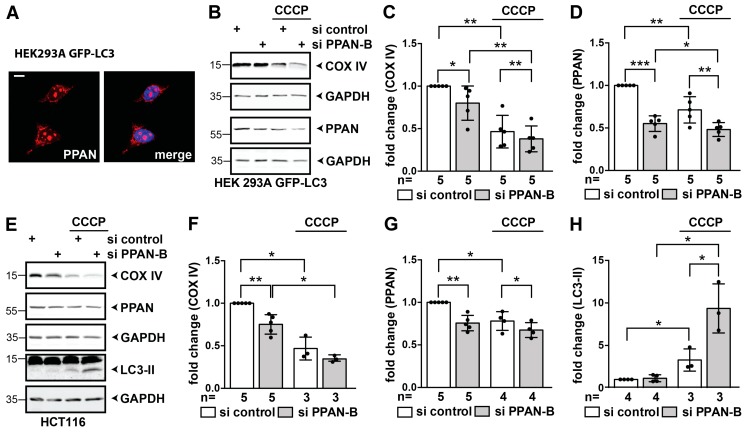
PPAN is decreased by CCCP and PPAN knockdown further sensitizes cells to mitochondrial loss. (**A**) Endogenous PPAN localizes to nucleoli and mitochondria in HEK293A GFP-LC3 cells. The confocal image is representative of 3 experiments. GFP-LC3 expressing cells (GFP fluorescence not shown) were stained with PPAN antibodies and DAPI (see merge). Scale bar, 10 µm. (**B**–**D**) CCCP-mediated activation of mitophagy significantly reduces COX IV and PPAN levels as monitored by Western blotting in HEK293A GFP-LC3 cells. PPAN knockdown in HEK293A GFP-LC3 cells as indicated above panels reveals decrease of COX IV upon CCCP treatment. In addition, PPAN is decreased following CCCP treatment (10 µM) for 24 h. Western blots of whole cell lysates probed with COX IV, PPAN and GAPDH antibodies. Lysates of the same experiment were loaded twice (see GAPDH placed below respective blots). (**C**) Quantification of COX IV normalized to GAPDH as shown in (B). Note that 4 out of 5 experiments were treated without DMSO (n = 4) and one experiment with DMSO. The si control in absence of CCCP was set to 1. (**D**) Quantification of PPAN normalized to GAPDH as shown in (**B**). The si control was set to 1. (**E**–**H**) PPAN knockdown in HCT116 cells as indicated above panels reveals decrease of COX IV, which is accompanied by increased LC3-II accumulation upon CCCP treatment. In addition, PPAN is decreased following CCCP treatment (20 µM) for 17 h. Controls were treated with DMSO as indicated. (**E**) Whole cell lysates were probed with COX IV, PPAN, GAPDH and total LC3 antibodies. Whole cell lysates of the same experiment were loaded twice. (**F**) Quantification of COX IV normalized to GAPDH as shown in (**E**) by densitometry. The DMSO treated si control cells were set to 1. (**G**) Quantification of PPAN normalized to GAPDH as shown in (**E**). The DMSO treated si control cells were set to 1. (**H**) Quantification of LC3-II normalized to GAPDH as shown in (**E**). The DMSO treated si control cells were set to 1. Error bars represent S.D., *, *p* < 0.05; **, *p* < 0.01; ***, *p* < 0.001; n = number of independent experiments.

**Figure 8 cells-08-00894-f008:**
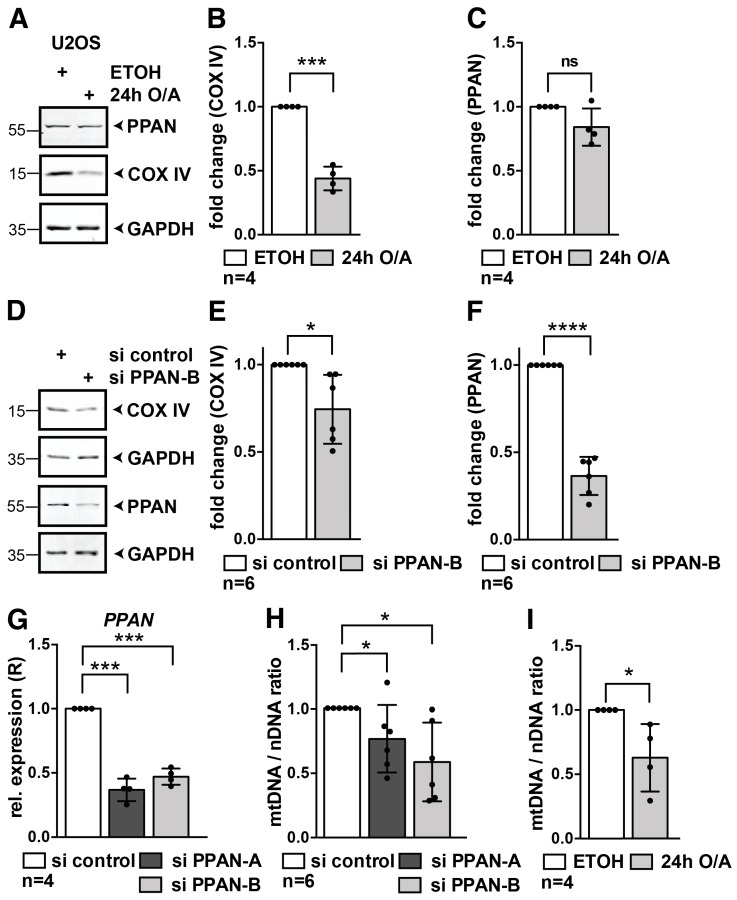
PPAN knockdown decreases mitochondrial mass and mtDNA to nuclear DNA ratio in U2OS cells. (**A**–**C**) COX IV and PPAN protein levels in wildtype U2OS cells upon 24 h O/A treatment as indicated. Controls were treated with EtOH (ethanol) as indicated. (**A**) Whole cell lysates were subjected to Western blotting. Membranes were probed with PPAN, COX IV and GAPDH antibodies. (**B**) Quantification of COX IV normalized to GAPDH as shown in (**A**). The EtOH treated cells were set to 1. (**C**) Quantification of PPAN normalized to GAPDH as shown in (**A**) by densitometry; ns, non-significant differences. (**D**–**F**) PPAN knockdown in U2OS cells decreases COX IV protein levels. (**D**) Whole cell lysates were subjected to Western blotting and membranes were probed with COX IV, PPAN and GAPDH antibodies. Lysates of the same experiment were loaded twice (see GAPDH placed below respective blots). (**E**) Quantification of COX IV normalized to GAPDH as shown in (**D**). (**F**) Quantification of PPAN normalized to GAPDH as shown in (**D**). The si control was set to 1. (**G**) qPCR analysis of U2OS cells transfected with control and PPAN siRNAs as indicated. Transfection of both PPAN siRNAs results in significant down-regulation of *PPAN* mRNA levels. *PPAN* was normalized to *GAPDH* (=relative expression, R). (**H**) Measurement of mtDNA/nDNA ratio by qPCR analysis of U2OS cells transfected with control and PPAN siRNAs as indicated. (**I**) Measurement of mtDNA/nDNA ratio of U2OS cells treated with EtOH or O/A for 24 h as indicated. The si control was set to 1. Error bars represent S.D.; *p* values were calculated by t-test. Statistically significant differences are indicated by asterisks, *, *p* < 0.05, ***, *p* < 0.001, ****, *p* < 0.0001; n = number of independent experiments.

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
