# Peer review of "Loss of Peter Pan (PPAN) Affects Mitochondrial Homeostasis and Autophagic Flux"

_cells, 2019, doi:10.3390/cells8080894_

Round 1
Reviewer 1 Report
The manuscript titled "Loss of Peter Pan (PPAN) affects mitochondrial architecture and autophagy flux" by Dannheisig et al., is very well presented with proper methods and logic. However, I felt disconnected often times with the flow of overall narration in Discussion Section. I think, Discussion can be more succinct and can be improved so that the take home message will be clear. Authors seem to have studied PPAN associated mechanisms and pathologies, however, in the reader's perspective, it can be better explained. Some minor corrections are highlighted in the attached draft.

Author Response
Please, see attached PDF with point-by-point-response.

Reviewer 2 Report
The manuscript entitled « Loss of Peter Pan affects mitochondrial architecture and autophagic flux » by Dannheisig D.P. et al. is quite interesting and fit well in and its interaction with cardiolipin and offered new incomes concerning the mitochondrial involvement in autophagy when its homeostasis is disrupted. At whole the manuscript is well, writen and quite complete.
At the overall this is a good manuscript but its length could be reduced since they are somehow some parts that are too long.
However, this manuscript has a very weak point. That is all the parts of the manuscript that is dealing with Peter Pan localization at the mitochondrial level. This could be justified by the fact that still few researchers that really know in detail the mitochondrial structures and functions in there full complexity. The main point concern cardiolipin accessibility to protein from the cytoplam (see the specific major remark below).
I would like to ask the authors to change the title to a more general one :
« Loss of Peter Pan affects mitochondrial homeostasis and autophagic flux ».
The term « homeaostasis » encompass the fact that mitochondrial functions and structures are affected by Peter Pan downregulation. Effectively, « mitochondrial architecture » is to much reductive when it is clear that the Peter Pan protein to CL interaction first disturbe the mitochondrial membrane structure (i.e. by its interaction with cardiolipin), the mitochondrial bioenergetic (as a consequence) and subsequently general mitochondria architecture. The term « homeostatsis » encompass all three domains.
Major remarks
At whole some parts of the manuscript could be shortened.
A special attention should be taken to avoid overlapping with previous report Pifster AS et al. (2015) to what concern mitochondria (Ref. 37, this manuscript).
The introdution is somehow to much diluted when autophagy is described line 52 – 65. Alos the lines 52-53 and 63-65 could be fusioned in one or two sentences.
• Page 2 line 44-45. The first sentence should be cut into two parts. The content should be more explicite.
• Page 3 line 88-89. what is the underlaying mechanism of PPAN shuttling from nucleus to mitochondria. If this has been described before in ref. 37, it will be nice to say some words here also. I just think that this is nuclear stress or initiation of apoptosis… that might be at work here.
Are the mitochondria redristibuted more or less close to the nuclear periphery ?
• Page 3 line 101. Suppress « present in the C-terminus of PPAN » (since this is written twice).
• Page 3 line 102. Could you please give us some informations about the sequence in term of amino-acids composition since there lipophilicity and charge (cationic) should be essential for mitochondrial targeting.
• Page 8 line 292. The sentence should say mitochondrial homeostasis instead of mitochondrial architecture and function.
• Page 8 part 3.1. It is quite difficult to follow this part not because it is complex.
But since your goal is clearly to demonstrate that PPAN interact at the mitochondrial level with CL, you may notice that CL located at the mitochondrial « contact sites » are disponible for initial binding and that mitochondrial membrane destabilization follows (from the literature it is wellknown that CL represent 18-20% of the contact site lipid composition). Thus after the first hit at the contact site, there is a redistribution of the CL within the membrane and the MIM resident CL are able to move to the OM and be available for any type of further interactions. When you perform a treatment with proteinase K you targeted proteins and not the CL associated to the membrane nor the contact sites that are special structure where the CL might into inverted structure into the membrane if calcium is present at the vinicity of the contact site which are the site where OM and IMM are in apposition (and that is the case). The model of protein import into the mitochondrial could be usefull but not totally relevant for proteins that indeed interacts with CL. So most of this part of the manuscript should be shortened and the discussion modified accordingly to these informations. At least for me the conclusion lines 337-338 is not true.
• Page 10 Lines 368-370 I agree for that but you may also say the PPAN almost acts as an antiapoptotic element by stabilizing mitochondrial outer membrane via its interaction with CL (an idea previously and partly introduced in ref. 37).
• Page 10 line 376. Here you introduced Pes1 as a partner of PPAN… this could have been done within the introduction. But, it is not important in the present manuscript.
*** Major point - Page 11 lines 388 – 390. Speaking about ATP, the authors should absolutely realized an HPLC determination and quantification of the three folowing elements ATP, ADP and AMP. This is of huge importance and especially the changes in ATP/ADP ration and the level of AMP (and the AMP/ATP that may increase) since AMP kinase is dependent of the relative AMP level.
These determination are of major importance, because an increase in AMP/ATP ratio might lead to the activation of the AMP kinase, a major regulator of the mitochondrial biogenesis via the PGC1-α pathway, thus accounting for the observed increase in mitochondrial mass (even if represented by smaller mitochondria). The best approach for these observation is also to quantify citrate synthase
• General question…
In the previous publication you say « we placed PPAN upstream of cytochrome c as it has a pivotal role in blocking BAX stabilization and BAX-mediated mitochondrial membrane permeabilization ».
My question is that caspase-8 induction and tBid delocalization migth be necessay for Bax delocalization but you did not say a single word about this well described pathway. Indeed, this is important since it has been demonstrated that tBid interact with CL as well as the activated caspase-8 do (Cardiolipin provides an essential activating platform for caspase-8 on mitochondria, Gonzalvez F. et al. 2008, J. Cell Biol. Vol. 183 No. 4, 681–696 www.jcb.org/cgi/doi/10.1083/jcb.200803129) and the subsequent events at the mitochondrial level (Cell Death and Differentiation (2005) 12, 614–626. doi:10.1038/sj.cdd.4401571), taken from a bioenergetic point of view (concerning caspase-8 and Bid) have been described (Gonzalvez F. et al. 2013) and that these mechanisms are widely accepted. You may have a look also to Jalmar et al. 2013 PLOS One. 8 (2): e55250. doi:10.1371/journal.pone.0055250). And concerning the Barth syndrome (Gonzalvez et al. 2013 Biochim Biophys Acta. 1832 (8) : 1194-1206. doi: 10.1016/j.bbadis.2013.03.005). Some of these articles clearly dealing with CL and mitochondrial bioenergetic can help the authors to understand the key mechanisms of what they have experimented and to draw a more detailled discussion. Citations of some of them could be useful.
• Page 17 instead of « si » put siRNA » in the text.
• Page 20 line 612… and Page 22 line 662. The measurement of mitochondrial mass is very problematic since fractionation of mitochondria do not really mean mitochondrial mass decrease at the cellular level. And it clear that the authors do not really tell us what they use as mitochondrial mass measurement… Mitochondria loss is another distinct concept espacially when mitophagy is at work. A measurement of mitochondrail DNA content by cell is not a proof of mitochondrial mass measurement.
• All parts of the manuscript and references that concerned sulphatides should be suppressed from the present manuscript.
Page 26… lines 799 – 805 the part of the manuscript concerning the sulfatides is presently irrelevant and should be discarbed from the present manuscript . Since this may concern another manuscript. Refs 80-83 should be deleted.
Minor remarks
Cytochrome c : should be writen cytochrome c (in italic) as its come from latin writing.
Author Response

(The authors gave the same response as above.)
